# Association between receiving the *Aksi Bergizi* Social Behavioral Change Communication (SBCC) intervention and dietary habits among secondary school students in Padang, Indonesia

Ricvan Dana Nindrea[1,2], Sawitri Assanangkornchai[1], Masrul Muchtar[3], Virasakdi Chongsuvivatwong[1], Wit Wichaidit[1]*

1 Department of Epidemiology, Faculty of Medicine, Prince of Songkla University,Hat Yai, Thailand,
2 Department of Medicine, Faculty of Medicine, Universitas Negeri Padang, Bukittinggi, Indonesia,
3 Department of Nutrition, Faculty of Medicine, Andalas University, Padang, Indonesia

* wit.w@psu.ac.th

## Abstract

### Background

The Government of Indonesia and UNICEF introduced the *Aksi Bergizi* Social Behavioral Change Communication (SBCC) intervention to promote healthy dietary behaviors among adolescents. However, no systematic assessment of the Program's effect has been made. The objectives of this study are: 1) to assess the extent to which exposure to the *Aksi Bergizi* Program is associated with dietary behaviors among secondary school students, and; 2) to assess mediation of the mentioned association by dietary self-efficacy.

### Methods

We conducted a school-based cross-sectional study in Padang Municipality, Indonesia. We collected data from 253 students attending *Aksi Bergizi* target schools, and 253 students from non-target schools, using self-administered questionnaire with a food frequency questionnaire (FFQ) section. We analyzed data using descriptive statistics with analysis of mediation by self-efficacy in dietary consumption.

### Results

We identified three distinct dietary patterns among the students: one characterized by higher frequencies of eating meat, processed foods, and dessert (i.e., "High Protein and Processed Foods" dietary pattern), another by higher frequencies of eating snacks and sweetened drinks ("Snacks and Sugary Drinks"), and another by higher frequencies of eating soybean products and fresh fruits and lower frequency of eating preserved vegetables ("Healthier Diet"). Students in the Aksi Bergizi target schools

**Data availability statement:** The data underlying the results presented in thie study are available in the Supporting Information section.

**Funding:** The first author (RDN) received financial support for data collection in this research work from the TUYF Charitable Trust: Research Capacity through Education and Networking on Epidemiology in Asia, the Department of Epidemiology, Faculty of Medicine, Prince of Songkla University (Grant number 1/2022). The funders had no role in study design, data collection and analysis, decision to publish, or preparation of the manuscript.

**Competing interests:** The authors have declared that no competing interests exist.

were significantly less likely than students in the non-target schools to have the High Protein and Processed Foods dietary pattern (40.2% vs. 53.9%, Adjusted OR = 0.44; 95% CI = 0.20, 0.99) and more likely to have a Healthier Diet pattern, although the difference of the latter was not statistically significant. Students in the target schools, however, were also more likely than students in the non-target schools to have the Snacks and Sugary Drinks dietary pattern (78.7% vs. 40.4%, Adjusted OR = 6.22; 95% CI = 2.68, 14.42). Regarding mediation, students in the Aksi Bergizi target schools had significantly different dietary self-efficacy from students in the non-target schools (p < 0.001). However, self-efficacy was not significantly associated with dietary habits. The association in the non-mediated pathway between exposure to the Aksi Bergizi program and dietary habits was statistically significant (p < 0.001).

## Conclusion

The findings of this study have implications for stakeholders in adolescent health. However, limitations regarding the cross-sectional design (which precludes the ability to make causal inference), social desirability, and limited generalizability should be considered in the interpretation of the study findings.

## Introduction

Adolescents face a double burden in malnutrition [1]. The World Health Organization (WHO) estimated that 19% of adolescents worldwide were stunted, 8% were underweight, 9% had wasting, 11% were overweight and 9% were obese [2]. Malnutrition in adolescents is partly attributed to dietary habits [3,4]. A meta-analysis study found that 35% of adolescents aged 12–17 years did not consume fruit at least 1 time/per day, 21% did not consume vegetables at least once per day, 43% consumed carbonated soft drinks at least once per day, and 46% consumed fast food at least once per week [4]. Unhealthy dietary habits during adolescence can elevate the risk of non-communicable diseases (NCDs) during adolescence and adulthood [5].

Indonesia, a middle-income country where approximately 16 percent of the population are adolescents [6], also faces problems in adolescent malnutrition, including lack of nutrients, obesity, and anemia [7,8]. To promote healthier dietary behavior among adolescents, the Government of Indonesia and the United Nations Children's Fund (UNICEF) have designed and implemented an intervention based on Social Behavioral Change Communication (SBCC) [7] in selected secondary schools nationwide known as the *Aksi Bergizi* Nutrition Promotion Program [9–11]. Although *Aksi Bergizi* has been implemented since the middle of 2022, no assessment has been made regarding the extent to which exposure to the Program's activities was associated with dietary behaviors. Furthermore, as Social Behavioral Change Communication's theory of change posits that changes occur through psychosocial drivers of the target behavior [12–14] particularly but not limited to self-efficacy [15–17]. In other words, Social Behavioral Change Communication interventions can increase levels of self-efficacy for target behaviors, which then increases the behaviors' likelihood.

Thus, we hypothesize that the effect of the intervention program on dietary behaviors is mediated by self-efficacy. Findings from such assessment can contribute empirical evidence regarding the program's effectiveness. Thus, the findings have implications for stakeholders in adolescent health for guiding future decisions regarding the Program's extension or further modification. The objectives of this study are: 1) to assess the extent to which exposure to the *Aksi Bergizi* Nutrition Promotion Program is associated with dietary behaviors among secondary school students in Padang, Indonesia, and; 2) to assess the extent to which the mentioned association is mediated by self-efficacy of dietary behaviors.

## Methods

### Study design and setting

We conducted a school-based cross-sectional study in Padang Municipality, West Sumatra Province, Indonesia. We collected data from classrooms in four government (public) schools in urban and peri-urban areas.

### Study participants and sample size calculation

The study participants were secondary school students aged 12–15 years in Padang Municipality, West Sumatra Province, Indonesia. For sample size calculation, we hypothesized that 59% of students in schools that received the *Aksi Bergizi* Program at fruit at least once per day ($p_1 = 0.59$) compared to 42% among students in schools without the *Aksi Bergizi* Program ($p_2 = 0.42$) [18,19]. Using the sample size calculation formula for comparison of two proportions and procedures for calculation using the R statistical environment [20] assuming 95% level of confidence and 80% power, we obtained the sample size of 253 students per group, thus a total of 506 students.

### The *Aksi Bergizi* nutrition promotion program

The aim of the *Aksi Bergizi* nutrition promotion program was to provide health education to adolescents aged 12–18 of all genders in the target schools through a series of activities based on Social Behavioral Change Communication (SBCC). Program activities included nutritional education and promotion of healthy eating habits. The theory of change was that by increasing adolescents' knowledge and awareness of good nutrition and a healthy lifestyle, the adolescents would be more likely to adopt healthy eating patterns. The Program tasked either the teachers at the school or nutrition staff from a nearby primary healthcare center (*puskesmas*) to provide health education and organize activities on the first week of every month. Details regarding the Program's intervention are available from a public source [9–11]. As the *Aksi Bergizi* Program was implemented in entire schools, we assumed that all students in schools that were the Program's target had been exposed to the Program.

### Measurement of dietary habits

We used the food frequency questionnaire built based on the items listed in a previous study on dietary behaviors in Indonesia to measure dietary habits among students in the participating schools [10,11]. The FFQ section consisted of 22 questions, focusing on food items that are localized or particular to Indonesia, with the following possible answer choices: never or less than once per month, 1–3 times per month, once a week, 2–4 times per week, 5–6 times per week, once a day, and more than once a day. We converted answers on the FFQ to binary responses (i.e., whether the participant ate a given food item more than once per week vs. once per week or fewer).

### Measurement of the mediator variable (self-efficacy)

The mediator variable was self-efficacy, which measured using the questions "*I am confident that I can limit my consumption of sugary drinks (e.g., soda, energy drinks) to one per week or less*" and "*I am confident that I can resist the temptation to engage in unhealthy foods (e.g., excessive snacking)*" [11]. The answers were in 4-point Likert scale with

 

the assigned value of 1 for "*Strongly disagree*", 2 for "*Disagree*", 3 for "*Agree*", and 4 for "*Strongly agree*". We added both variables into the structured equation model as they appeared.

## Study instrument

Our study instrument was a self-administered paper and pen questionnaire. We designed the questionnaire in English and Bahasa Indonesia by adapting existing instruments [10,11]. We conducted a pilot-test of the study instrument in Solok Regency, which is one of the regencies/cities in West Sumatra Province. Further revision of the questionnaire was carried out through an iterative process. We also assessed the validity and reliability of the instruments during this period. These variables were assessed using a Likert scale. The questions we used underwent validity and reliability testing, resulting in a Cronbach's alpha value of >0.7. The final questionnaire contained 3 sections and took approximately 20 minutes to complete.

## Sampling procedures

The secondary school database was obtained from the Padang Municipality Education Office, West Sumatra Province, Indonesia. The sampling technique used in this study was multistage stratified clustered sampling. In the first stage, the secondary school database was obtained from the Padang Municipality Education Office, West Sumatra Province, Indonesia. A stratum was created based on secondary schools being targeted and non-targeted for the *Aksi Bergizi* nutrition program. Two schools were then randomly selected from each group. There were four secondary schools (all public) that were targeted for the *Aksi Bergizi* nutrition program and 39 secondary schools that were not targeted for the program. Two secondary schools were recruited from each group. In each school, we performed stratified random sampling of classrooms to select two classrooms per grade level using the list of classrooms provided by the school. To avoid any potential repercussions, we hereby refrain from disclosing the names of the schools.

## Data collection

We scheduled an appointment with the selected secondary schools to determine a feasible time and date for data collection in the selected classrooms. We then asked the principal or the teacher in charge to introduce the investigators to the students in the classrooms. The investigators informed the students about the study and requested their verbal informed consent. We requested and obtained a waiver of written informed consent and parental consent from our Institutional Review Boards (IRBs) in order to help us reassume the participants of their confidentiality. The information and consent processes were conducted in a group setting within the selected classes, but we allowed for individual questions and answers during the recruitment process. After the students expressed verbal consent, we directly distributed the study questionnaires to the students and asked them to start filling out the questionnaires. We also ensured that no teacher was present in the classroom at the time. At the end of the data collection period, the students placed their questionnaire in an opaque envelope provided by the investigators, and placed the envelope in a locked box in front of the classroom or at a location otherwise designated by the investigators. Data collection started on 9 November 2023 and ended on 2 December 2023.

## Data management

We opened the secured box in a private location and performed data entry on the paper questionnaires using the KoboToolbox platform, which uploaded the entered data to a password-protected server. For each questionnaire, two investigators performed data entry separately, and the principal investigator (RDN) checked for discrepancies between the two versions with regard to the unique identification number (ID) and other values, checked the original questionnaire, and made corrections accordingly.

## Data analysis

We used descriptive statistics to describe the characteristics of the study participants. We conducted bivariate analysis using the Chi-square test and multivariate logistic regression analysis, adjusting for potential confounders based on demographic characteristics. We performed exploratory factor analyses to identify dietary patterns. We decided on the appropriate number of patterns based on the Scree plot and a consensus among the investigators who performed data analyses. We then used the 75th percentile ranking of each dietary pattern as an arbitrary cut-off point for having such a dietary pattern, and cross-tabulated the presence of a dietary pattern with the binary consumption of each food item to validate factor analysis results. Structural equation modelling (SEM) was used to analyze the data. A confirmatory factor analysis (CFA) was employed to test the measurement model of the hypothesized model. A p-value of <0.05 was considered statistically significant.

## Ethical considerations

This study received ethical approval from the ethics committee of the Faculty of Medicine at Prince Songkla University (Approval No. REC.66-248-18-2).

## Results

A total of 253 students from the *Aksi Bergizi* target schools and 253 students from the *Aksi Bergizi* non-target schools participated in our study. There were significant socio-demographic differences between participants target and non-target schools (Table 1). While no significant differences were found in sex, religion, and BMI distributions, notable variations were observed. Targeted schools had a higher proportion of 14-year-old participants, fathers with an associate's degree, and households earning 3–4 million IDR monthly. Conversely, non-target schools had a higher percentage of fathers working as laborers/manual workers, mothers working as housewives, and mothers with a junior high school education. Additionally, non-targeted schools had more participants citing family and/or friends as their primary source of health behavior information. Students in the *Aksi Bergizi* target schools were less likely to self-report self-efficacy in maintaining a healthy diet, such as limiting the consumption of sugary drinks and excessive snacking.

Prevalence of dietary habits among students exposed to the *Aksi Bergizi* program (i.e., students at target school) and students not exposed to the program (i.e., students at non-target school) are as shown in Table 2. Frequency of consumption differed significantly between students at *Aksi Bergizi* target schools and non-target schools except for that of white rice and brown rice. Overall, students at *Aksi Bergizi* target schools tended to have a higher frequency of eating healthier foods, such as fresh fruits. However, students at target schools also had a significantly higher proportion of those who ate deep-fried fish or meat almost daily.

We performed exploratory factor analyses for the food items, the Scree plot suggested two potential cut-off points (either at 2 factors or 4 factors) (Fig 1). After internal deliberation, the authors decided on cutting at 3 factors. The first factor loaded on 45% of variations in food consumption frequency (proportion explained by PA1 = 0.45), the second factor loaded on 31% (proportion explained by PA2 = 0.31, and the third factor loaded on the remaining 24% (proportion explained by PA3 = 0.24) (Table 3). Based on the factor loadings correlation patterns, we decided to name the first factor the "Dietary Pattern 1: High Protein and Processed Foods", the second factor the "Dietary Pattern 2: Snacks and Sugary Drinks", and the third factor the "Dietary Pattern 3: Healthier Diet". Validation of factor analysis results by cross-tabulation between having dietary patterns identified by factor analysis by frequency of consumption of individual food items is shown in S1 Table.

Students in the *Aksi Bergizi* target schools were significantly less likely than students in the non-target schools to have the High Protein and Processed Food dietary pattern (40.2% vs. 53.9%, Adjusted OR = 0.44; 95% CI 0.20, 0.99) (Table 4). Students in the target schools were more likely than students in the non-target schools to have a Healthier Diet

**Table 1. Characteristics of the study participants in *Aksi Bergizi* target and non-target schools (Frequency and %).**

| Characteristic | *Aksi Bergizi* target schools (n = 253 students) | *Aksi Bergizi* non-target schools (n = 253 students) | P-value |
|---|---|---|---|
| **Sex** | | | |
| Male | 106 (41.9%) | 111 (43.9%) | 0.719 |
| Female | 147 (58.1%) | 142 (56.1%) | |
| **Age (years)** | | | |
| 12 | 12 (4.7%) | 22 (8.7%) | **0.036** |
| 13 | 72 (28.5%) | 90 (35.6%) | |
| 14 | 115 (45.5%) | 88 (34.8%) | |
| 15 | 54 (21.3%) | 53 (20.9%) | |
| **Ethnicity** | | | |
| Minangnese | 146 (57.7%) | 167 (66.0%) | **<0.001** |
| Javanese | 35 (13.8%) | 57 (22.5%) | |
| Bataknese | 8 (3.2%) | 0 (0%) | |
| Sundanese | 5 (2.0%) | 0 (0%) | |
| Others | 59 (23.3%) | 29 (11.5%) | |
| **Father's occupation** | | | |
| Civil servant/ state enterprise | 54 (21.3%) | 74 (29.2%) | **<0.001** |
| Private sector employee | 92 (36.4%) | 79 (31.2%) | |
| Small-scale vendors/ service providers | 42 (16.6%) | 30 (11.9%) | |
| Business owner/ entrepreneur | 26 (10.3%) | 10 (4.0%) | |
| Laborer/ manual workers | 18 (7.1%) | 60 (23.7%) | |
| Agriculture/ fishery | 19 (7.5%) | 0 (0%) | |
| Independent professions (e.g., lawyers, architects) | 2 (0.8%) | 0 (0%) | |
| **Father's education** | | | |
| Junior high school | 38 (15.0%) | 65 (25.7%) | **<0.001** |
| Senior high school | 86 (34.0%) | 93 (36.8%) | |
| Vocational certificate | 1 (0.4%) | 0 (0%) | |
| Associate's degree | 70 (27.7%) | 26 (10.3%) | |
| Bachelor's degree | 51 (20.2%) | 69 (27.3%) | |
| Higher than bachelor's degree | 7 (2.8%) | 0 (0%) | |
| **Mother's occupation** | | | |
| Housewife | 173 (68.4%) | 199 (78.7%) | **0.036** |
| Civil servant/ state enterprise | 15 (5.9%) | 12 (4.7%) | |
| Private sector employee | 22 (8.7%) | 12 (4.7%) | |
| Small-scale vendors/ service providers | 39 (15.4%) | 30 (11.9%) | |
| Business owner/ entrepreneur | 4 (1.6%) | 0 (0%) | |
| **Mother's education** | | | |
| Primary school | 2 (0.8%) | 0 (0%) | **0.031** |
| Junior high school | 45 (17.8%) | 63 (24.9%) | |
| Senior high school | 101 (39.9%) | 101 (39.9%) | |
| Vocational certificate | 4 (1.6%) | 10 (4.0%) | |
| Associate's degree | 80 (31.6%) | 69 (27.3%) | |
| Bachelor's degree | 21 (8.3%) | 10 (4.0%) | |
| **Household monthly income** | | | |
| <1,000,000 IDR | 0 (0%) | 0 (0%) | **<0.001** |
| 1,000,000–2,000,000 IDR | 31 (12.3%) | 66 (26.1%) | |

*(Continued)*

**Table 1.** (Continued)

| Characteristic | *Aksi Bergizi* target schools (n = 253 students) | *Aksi Bergizi* non-target schools (n = 253 students) | P-value |
|---|---|---|---|
| 2,000,001–3,000,000 IDR | 61 (24.1%) | 77 (30.4%) | |
| 3,000,001–4,000,000 IDR | 95 (37.5%) | 72 (28.5%) | |
| 4,000,001–5,000,000 IDR | 47 (18.6%) | 33 (13.0%) | |
| 5,000,001–6,000,000 IDR | 19 (7.5%) | 5 (2.0%) | |
| **Religion** | | | |
| Islam | 240 (94.9%) | 240 (94.9%) | 0.999 |
| Christianity | 13 (5.1%) | 13 (5.1%) | |
| **Body mass index (BMI)** | | | |
| Underweight (BMI < 18.5 kg/m$^2$) | 3 (1.2%) | 2 (0.8%) | 0.145 |
| Normal (BMI 18.5–22.9 kg/m$^2$) | 245 (96.8%) | 238 (94.1%) | |
| Overweight (23–24.9 kg/m$^2$) | 5 (2.0%) | 13 (5.1%) | |
| Obesity (≥25 kg/m$^2$) | 0 (0%) | 0 (0%) | |
| **Main sources of information about health behaviors** | | | |
| Television advertisements | 63 (24.9%) | 41 (16.2%) | **0.003** |
| Family and/or friends | 131 (51.8%) | 169 (66.8%) | |
| Social media platforms (e.g., Facebook, X, Instagram, etc) | 59 (23.3%) | 43 (17.0%) | |
| ***Dietary self-efficacy*** | | | |
| **Agree or strongly agree:** I am confident that I can limit my consumption of sugary drinks (e.g., soda, energy drinks) to one per week or less | 253 (100.0%) | 207 (81.8%) | **<0.001** |
| **Agree or strongly agree:** I am confident that I can resist the temptation to engage in unhealthy food (e.g., excessive snacking) | 253 (100.0%) | 228 (90.1%) | **<0.001** |

pattern (66.9% vs. 43.8%, Adjusted OR = 1.73; 95% CI 0.86, 3.47). Students in the target schools, however, were also more likely than the non-target schools to have the Snacks and Sugary Drinks dietary pattern Poultry and Snacks dietary pattern (78.7% vs. 40.4%, Adjusted OR = 6.22; 95% CI 2.68, 14.42).

Mediation of the association between exposure to *Aksi Bergizi* and dietary habits (Fig 2) showed that being in Aksi Bergizi target schools (i.e., exposure to the *Aksi Bergizi* program) was significantly associated with self-efficacy (P < 0.001). However, self-efficacy was not significantly associated with dietary habits. The non-mediated pathway between exposure to the *Aksi Bergizi* program and dietary habits was statistically significant (P < 0.001).

## Discussion

In this school-based cross-sectional study, we described the prevalence of self-reported health behaviors among students exposed to the *Aksi Bergizi* program and students not exposed to the program. We identified three distinct dietary patterns among the students: High Protein and Processed Foods, Snacks and Sugary Drinks, and Healthier Diet patterns. Students in *Aksi Bergizi* target schools were less likely than students in non-target schools to have the High Protein and Processed Foods dietary pattern and more likely to adopt the Healthier Diet pattern, although the latter association was not statistically significant. However, students in the target schools were also significantly more likely to have the Snacks and Sugary Drinks dietary pattern. The findings of our study have implications for stakeholders in adolescent health and school health programs.

Previous studies have highlighted the importance of nutrition education programs in promoting healthy dietary habits among adolescents [21–23]. However, in this study, we measured dietary behaviors only with the Food Frequency Questionnaire (FFQ), which did not measure the food items' composition. In other words, there was no data with regard to food

**Table 2. Frequency of food consumption by item among students exposed to the *Aksi Bergizi* program and students not exposed to the program (frequency and %).**

| Food item | *Aksi Bergizi* target schools (n = 253 students) | *Aksi Bergizi* non-target schools (n = 253 students) | P-value |
|---|---|---|---|
| **Rice (white rice)** | | | 0.285 |
| Never or less than once per month | 0 (0%) | 0 (0%) | |
| 1–3 times per month | 0 (0%) | 0 (0%) | |
| Once a week | 1 (0.4%) | 0 (0%) | |
| 2–4 times per week | 1 (0.4%) | 0 (0%) | |
| 5–6 times per week | 0 (0%) | 1 (0.4%) | |
| Once a day | 2 (0.8%) | 0 (0%) | |
| More than once a day | 249 (98.4%) | 252 (99.6%) | |
| **Refined wheat products (white bread, noodles)** | | | **0.001** |
| Never or less than once per month | 4 (1.6%) | 1 (0.4%) | |
| 1–3 times per month | 16 (6.3%) | 10 (4.0%) | |
| Once a week | 58 (22.9%) | 48 (19.0%) | |
| 2–4 times per week | 162 (64.0%) | 194 (76.7%) | |
| 5–6 times per week | 12 (4.7%) | 0 (0%) | |
| Once a day | 1 (0.4%) | 0 (0%) | |
| More than once a day | 0 (0%) | 0 (0%) | |
| **Coarse grain (brown rice)** | | | 0.059 |
| Never or less than once per month | 245 (96.8%) | 252 (99.6%) | |
| 1–3 times per month | 6 (2.4%) | 1 (0.4%) | |
| Once a week | 2 (0.8%) | 0 (0%) | |
| 2–4 times per week | 0 (0%) | 0 (0%) | |
| 5–6 times per week | 0 (0%) | 0 (0%) | |
| Once a day | 0 (0%) | 0 (0%) | |
| More than once a day | 0 (0%) | 0 (0%) | |
| **Whole grain wheat products (e.g., brown bread, whole wheat noodles)** | | | **0.001** |
| Never or less than once per month | 197 (77.9%) | 218 (86.2%) | |
| 1–3 times per month | 34 (13.4%) | 8 (3.2%) | |
| Once a week | 15 (5.9%) | 20 (7.9%) | |
| 2–4 times per week | 7 (2.8%) | 6 (2.4%) | |
| 5–6 times per week | 0 (0%) | 0 (0%) | |
| Once a day | 0 (0%) | 0 (0%) | |
| More than once a day | 0 (0%) | 1 (0.4%) | |
| **Tubers (cassava, taro, white yams, white potato)** | | | **<0.001** |
| Never or less than once per month | 1 (0.4%) | 1 (0.4%) | |
| 1–3 times per month | 6 (2.4%) | 1 (0.4%) | |
| Once a week | 21 (8.3%) | 17 (6.7%) | |
| 2–4 times per week | 204 (80.6%) | 234 (92.5%) | |
| 5–6 times per week | 21 (8.3%) | 0 (0%) | |
| Once a day | 0 (0%) | 0 (0%) | |
| More than once a day | 0 (0%) | 0 (0%) | |
| **Meat (beef, mutton)** | | | **0.002** |
| Never or less than once per month | 0 (0%) | 0 (0%) | |
| 1–3 times per month | 179 (70.8%) | 146 (57.7%) | |
| Once a week | 61 (24.1%) | 93 (36.8%) | |

*(Continued)*

**Table 2.** (Continued)

| Food item | *Aksi Bergizi* target schools (n = 253 students) | *Aksi Bergizi* non-target schools (n = 253 students) | P-value |
|---|---|---|---|
| 2–4 times per week | 13 (5.1%) | 10 (4.0%) | |
| 5–6 times per week | 0 (0%) | 0 (0%) | |
| Once a day | 0 (0%) | 4 (1.6%) | |
| More than once a day | 0 (0%) | 0 (0%) | |
| **Poultry (duck, chicken)** | | | **<0.001** |
| Never or less than once per month | 0 (0%) | 0 (0%) | |
| 1–3 times per month | 23 (9.1%) | 40 (15.8%) | |
| Once a week | 128 (50.6%) | 162 (64.0%) | |
| 2–4 times per week | 102 (40.3%) | 35 (13.8) | |
| 5–6 times per week | 0 (0%) | 0 (0%) | |
| Once a day | 0 (0%) | 15 (5.9%) | |
| More than once a day | 0 (0%) | 1 (0.4%) | |
| **Fish (raw, grilled, soup, not deep-fried)** | | | **<0.001** |
| Never or less than once per month | 21 (8.3%) | 27 (10.7%) | |
| 1–3 times per month | 131 (51.8%) | 26 (10.3%) | |
| Once a week | 92 (36.4%) | 154 (60.9%) | |
| 2–4 times per week | 9 (3.6%) | 31 (12.3%) | |
| 5–6 times per week | 0 (0%) | 0 (0%) | |
| Once a day | 0 (0%) | 12 (4.7%) | |
| More than once a day | 0 (0%) | 3 (1.2%) | |
| **Fresh seafood (e.g., clams, prawns, crabs, octopus)** | | | **<0.001** |
| Never or less than once per month | 94 (37.2%) | 0 (0%) | |
| 1–3 times per month | 92 (36.4%) | 105 (41.5%) | |
| Once a week | 60 (23.7%) | 127 (50.2%) | |
| 2–4 times per week | 7 (2.8%) | 11 (4.3%) | |
| 5–6 times per week | 0 (0%) | 0 (0%) | |
| Once a day | 0 (0%) | 10 (4.0%) | |
| More than once a day | 0 (0%) | 0 (0%) | |
| **Eggs** | | | **0.006** |
| Never or less than once per month | 0 (0%) | 0 (0%) | |
| 1–3 times per month | 1 (0.4%) | 2 (0.8%) | |
| Once a week | 1 (0.4%) | 4 (1.6%) | |
| 2–4 times per week | 237 (93.7%) | 245 (96.8%) | |
| 5–6 times per week | 12 (4.7%) | 0 (0%) | |
| Once a day | 0 (0%) | 0 (0%) | |
| More than once a day | 2 (0.8%) | 2 (0.8%) | |
| **Leafy green vegetables (e.g., Chinese cabbage, long bean, kale, spinach, yu choy, cucumber)** | | | **0.042** |
| Never or less than once per month | 2 (0.8%) | 5 (2.0%) | |
| 1–3 times per month | 0 (0%) | 0 (0%) | |
| Once a week | 0 (0%) | 3 (1.2%) | |
| 2–4 times per week | 2 (0.8%) | 4 (1.6%) | |
| 5–6 times per week | 0 (0%) | 0 (0%) | |
| Once a day | 48 (19.0%) | 29 (11.5%) | |
| More than once a day | 201 (79.4%) | 212 (83.8%) | |

*(Continued)*

**Table 2.** (Continued)

| Food item | Aksi Bergizi target schools (n = 253 students) | Aksi Bergizi non-target schools (n = 253 students) | P-value |
|---|---|---|---|
| **Yellow or orange vegetables (e.g., pumpkin, sweet potatoes, carrots, ripened papaya)** | | | **0.004** |
| Never or less than once per month | 0 (0%) | 0 (0%) | |
| 1–3 times per month | 1 (0.4%) | 0 (0%) | |
| Once a week | 7 (2.8%) | 15 (5.9%) | |
| 2–4 times per week | 219 (86.6%) | 230 (90.9%) | |
| 5–6 times per week | 7 (2.8%) | 0 (0%) | |
| Once a day | 15 (5.9%) | 8 (3.2%) | |
| More than once a day | 4 (1.6%) | 0 (0%) | |
| **Soybean products (e.g., tofu, _tempe_)** | | | **<0.001** |
| Never or less than once per month | 2 (0.8%) | 3 (1.2%) | |
| 1–3 times per month | 0 (0%) | 0 (0%) | |
| Once a week | 7 (2.8%) | 24 (9.5%) | |
| 2–4 times per week | 181 (71.5%) | 221 (87.4%) | |
| 5–6 times per week | 60 (23.7%) | 5 (2.0%) | |
| Once a day | 1 (0.4%) | 0 (0%) | |
| More than once a day | 2 (0.8%) | 0 (0%) | |
| **Preserved vegetables (e.g., canned pickled vegetables)** | | | **0.008** |
| Never or less than once per month | 244 (96.4%) | 244 (96.4%) | |
| 1–3 times per month | 7 (2.8%) | 0 (0%) | |
| Once a week | 0 (0%) | 2 (0.8%) | |
| 2–4 times per week | 2 (0.8%) | 7 (2.8%) | |
| 5–6 times per week | 0 (0%) | 0 (0%) | |
| Once a day | 0 (0%) | 0 (0%) | |
| More than once a day | 0 (0%) | 0 (0%) | |
| **Fresh fruits** | | | **<0.001** |
| Never or less than once per month | 3 (1.2%) | 9 (3.6%) | |
| 1–3 times per month | 1 (0.4%) | 5 (2.0%) | |
| Once a week | 36 (14.2%) | 53 (20.9%) | |
| 2–4 times per week | 183 (72.3%) | 179 (70.8%) | |
| 5–6 times per week | 8 (3.2%) | 0 (0%) | |
| Once a day | 11 (4.3%) | 7 (2.8%) | |
| More than once a day | 11 (4.3%) | 0 (0%) | |
| **Dairy products (fresh milk, powdered milk, boxed milk)** | | | **<0.001** |
| Never or less than once per month | 163 (64.4%) | 203 (80.2%) | |
| 1–3 times per month | 7 (2.8%) | 5 (2.0%) | |
| Once a week | 4 (1.6%) | 10 (4.0%) | |
| 2–4 times per week | 22 (8.7%) | 5 (2.0%) | |
| 5–6 times per week | 6 (2.4%) | 0 (0%) | |
| Once a day | 50 (19.8%) | 29 (11.5%) | |
| More than once a day | 1 (0.4%) | 1 (0.4%) | |
| **Packaged snacks (e.g., corn puffs, Lay potato chips)** | | | **<0.001** |
| Never or less than once per month | 2 (0.8%) | 15 (5.9%) | |
| 1–3 times per month | 50 (19.8%) | 50 (19.8%) | |
| Once a week | 73 (28.9%) | 115 (45.5%) | |

_(Continued)_

| Food item | *Aksi Bergizi* target schools (n = 253 students) | *Aksi Bergizi* non-target schools (n = 253 students) | P-value |
|---|---|---|---|
| 2–4 times per week | 120 (47.4%) | 67 (26.5%) | |
| 5–6 times per week | 1 (0.4%) | 0 (0%) | |
| Once a day | 5 (2.0%) | 6 (2.4%) | |
| More than once a day | 2 (0.8%) | 0 (0%) | |
| **Unpackaged snacks (donuts, french fries,    roti, etc.)** | | | **<0.001** |
| Never or less than once per month | 1 (0.4%) | 0 (0%) | |
| 1–3 times per month | 53 (20.9%) | 61 (24.1%) | |
| Once a week | 87 (34.4%) | 114 (45.1%) | |
| 2–4 times per week | 108 (42.7%) | 59 (23.3%) | |
| 5–6 times per week | 0 (0%) | 14 (5.5%) | |
| Once a day | 2 (0.8%) | 5 (2.0%) | |
| More than once a day | 2 (0.8%) | 0 (0%) | |
| **Sweetened drinks or condiments (soy milk, soft drinks, coke, coffee or tea with sugar, sweetened condensed milk)** | | | **<0.001** |
| Never or less than once per month | 18 (7.1%) | 0 (0%) | |
| 1–3 times per month | 39 (15.4%) | 69 (27.3%) | |
| Once a week | 55 (21.7%) | 66 (26.1%) | |
| 2–4 times per week | 130 (51.4%) | 113 (44.7%) | |
| 5–6 times per week | 5 (2.0%) | 0 (0%) | |
| Once a day | 5 (2.0%) | 5 (2.0%) | |
| More than once a day | 1 (0.4%) | 0 (0%) | |
| **Processed or ultra-processed foods (canned fish, sausages, canned vegetables, frozen food, etc.)** | | | **<0.001** |
| Never or less than once per month | 52 (20.6%) | 5 (2.0%) | |
| 1–3 times per month | 145 (57.3%) | 154 (60.9%) | |
| Once a week | 39 (15.4%) | 65 (25.7%) | |
| 2–4 times per week | 16 (6.3%) | 29 (11.5%) | |
| 5–6 times per week | 0 (0%) | 0 (0%) | |
| Once a day | 1 (0.4%) | 0 (0%) | |
| More than once a day | 0 (0%) | 0 (0%) | |
| **Dessert (e.g., ice cream, cake, candy, cookies, other sweets)** | | | **0.005** |
| Never or less than once per month | 2 (0.8%) | 1 (0.4%) | |
| 1–3 times per month | 173 (68.4%) | 137 (54.2%) | |
| Once a week | 41 (16.2%) | 74 (29.2%) | |
| 2–4 times per week | 36 (14.2%) | 38 (15.0%) | |
| 5–6 times per week | 0 (0%) | 0 (0%) | |
| Once a day | 1 (0.4%) | 3 (1.2%) | |
| More than once a day | 0 (0%) | 0 (0%) | |
| **Deep-fried meat or fish (fried, fish, fried chicken, fried beef)** | | | **<0.001** |
| Never or less than once per month | 0 (0%) | 0 (0%) | |
| 1–3 times per month | 2 (0.8%) | 0 (0%) | |
| Once a week | 6 (2.4%) | 16 (6.3%) | |
| 2–4 times per week | 165 (65.2%) | 227 (89.7%) | |
| 5–6 times per week | 80 (31.6%) | 10 (4.0%) | |
| Once a day | 0 (0%) | 0 (0%) | |
| More than once a day | 0 (0%) | 0 (0%) | |

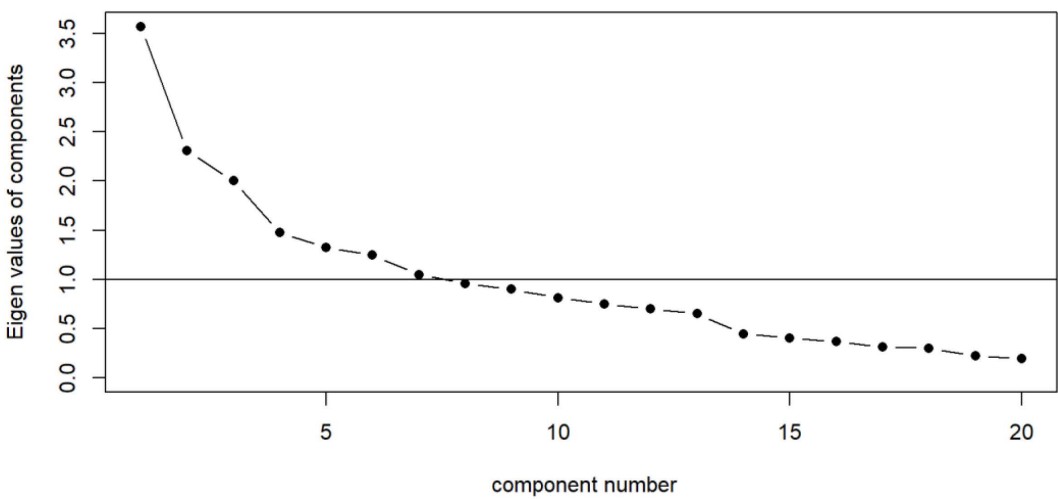

**Fig 1. Scree plot of dietary patterns classification among the participants.**

quantity, safety, or salt and sugar contents, all of which can impact health. In Padang, a typical meal often includes a staple carbohydrate such as rice accompanied by a variety of side dishes that include meat (fish, chicken, beef), vegetables, and *sambal* (chili paste) [24,25]. Rice often comprise more than 50% of daily caloric intake [26,27], and traditional Padang cuisine has liberal use of salt, sugar, and oil [27], as well as coconut milk. Thus, it is possible that there was excessive intake of salt, sugar, and fat independent of food frequency patterns [26,27]. Furthermore, in our analyses, we dichotomized the responses to the food frequency data as the original answers were ordinal variables with unequal intervals between the categories. Including the responses as integers could have introduced errors during exploratory factor analysis. Thus, we decided to collapse the responses into binary classifications. Such a practice is common in factor analysis [28] and could help to yield clear dietary patterns. However, one key limitation of this process is the loss of details, and the dichotomization as more than once per week vs. once per week or less also implies that there was no seasonality in the consumption of the food items. In the study region, families may consume meat only once per year during religious festivals, but do so in considerable quantity each day for up to one week (e.g., during Eid al-Adha and Eid al-Fitr) [24,29,30], yet the dichotomization presented in our study would not have captured this habit. Incorporating other measurement methods to capture these variations, such as the 24-hour food recall recorded throughout the year, could yield additional insights and enable more detailed measurements.

We found that students in *Aksi Bergizi* target schools were significantly more likely to adopt the Snacks and Sugary Drinks dietary pattern than students in non-target schools. While the *Aksi Bergizi* program aimed to improve dietary habits, these results indicate that the program's effects on dietary choices might have been contrary to expectations. This unexpected change was not found in a similar study in another region of Indonesia [31]. Another study showed that a school in Indonesia with on-site food service with a nutritionist had better nutrient adequacy among students than a school that had catering without a nutritionist [32]. Thus, it is possible that the lack of a catering component with nutritionist involvement could explain for parts of the observed unhealthy dietary patterns in our study. Regarding mediation analysis, exposure to the *Aksi Bergizi* program was significantly associated with improved dietary self-efficacy. However, self-efficacy was not significantly linked to dietary habits, indicating that other behavioral determinants might be more relevant in influencing students' eating patterns [12–14]. This suggests that while *Aksi Bergizi* successfully influenced students' confidence in making healthier dietary choices, it did not necessarily translate into consistent behavior change.

**Table 3. Factor loading for exploratory factor analysis of food consumption patterns among the study participants (n = 506 students).**

| Food item | PA1 ("High Protein & Processed Foods") | PA2 ("Snacks & Sugary Drinks") | PA3 ("Healthier Diet") |
|---|---|---|---|
| Rice (white rice) | −0.09 | 0.12 | 0.26 |
| Refined wheat products (white bread, noodles) | 0.05 | 0.27 | 0.10 |
| Coarse grain (brown rice) | | | |
| Whole grain wheat products (e.g., brown bread, whole wheat noodles) | 0.31 | 0.12 | −0.10 |
| Tubers (cassava, taro, white yams, white potato) | −0.05 | 0.24 | 0.14 |
| Meat (beef, mutton) | **0.71*** | 0.03 | −0.04 |
| Poultry (duck, chicken) | 0.32 | 0.22 | 0.00 |
| Fish (raw, grilled, soup, not deep-fried) | **0.67*** | −0.19 | 0.02 |
| Fresh seafood (e.g., clams, prawns, crabs, octopus) | **0.77*** | −0.34 | −0.13 |
| Eggs | 0.11 | 0.00 | −0.30 |
| Leafy green vegetables (e.g., Chinese cabbage, long bean, kale, spinach, yu choy, cucumber) | −0.24 | 0.00 | 0.31 |
| Yellow or orange vegetables (e.g., pumpkin, sweet potatoes, carrots, ripened papaya) | 0.19 | 0.16 | 0.19 |
| Soybean products (e.g., tofu, tempe) | −0.16 | 0.28 | **0.55*** |
| Preserved vegetables (e.g., canned pickled vegetables) | −0.03 | 0.16 | **−0.60*** |
| Fresh fruits | 0.26 | 0.01 | **0.57*** |
| Dairy products (fresh milk, powdered milk, boxed milk) | 0.37 | 0.38 | 0.00 |
| Packaged snacks (e.g., corn puffs, Lay potato chips) | 0.19 | **0.54*** | 0.11 |
| Unpackaged snacks (donuts, french fries, roti, etc.) | 0.14 | **0.78*** | −0.27 |
| Sweetened drinks or condiments (soy milk, soft drinks, coke, coffee or tea with sugar, sweetened condensed milk) | 0.17 | **0.48*** | −0.20 |
| Processed or ultra-processed foods (canned fish, sausages, canned vegetables, frozen food, etc.) | **0.54*** | 0.19 | −0.17 |
| Dessert (e.g., ice cream, cake, candy, cookies, other sweets) | **0.58*** | 0.19 | −0.03 |
| Deep-fried meat or fish (fried, fish, fried chicken, fried beef) | −0.07 | 0.26 | 0.05 |
| **SS loadings** | 2.82 | 1.90 | 1.49 |
| **Proportion Var** | 0.13 | 0.09 | 0.07 |
| **Cumulative Var** | 0.13 | 0.22 | 0.30 |
| **Proportion Explained** | 0.45 | 0.31 | 0.24 |
| **Cumulative Proportion** | 0.45 | 0.76 | 1.00 |

*Denotes the variable being able to explain variations beyond the threshold (rho > 0.40 or rho < −0.40)

The lack of association between self-efficacy and dietary habits in our SEM thus challenges our proposed theoretical model, and suggests the need to explore other potential mediators in the association between SBCC intervention and dietary behaviors.

The strength of this study lies in the self-administration of the study questionnaire, which helps to reduce social desirability bias, to an extent. However, a number of limitations should be considered in the interpretation of our study findings. Firstly, the cross-sectional design only allows for the description of variations in risky health behaviors at a single point in time. Our participants may engage in more or fewer risky behaviors during their life course. Secondly, with standard curriculum in health education, self-report of eating unhealthy foods could be regarded as not socially desirable. Thus, we could not rule out the influence of social desirability in our study findings. Lastly, we only collected data from schools in Padang Municipality, which limited the generalizability of our study findings.

**Table 4. Cross-tabulate dietary patterns found among students exposed to the *Aksi Bergizi* program and students not exposed to the program (row percent).**

| Dietary patterns | Pattern not present | Pattern present | Unadjusted OR (95% CI) | Adjusted OR (95% CI)* |
|---|---|---|---|---|
| **Dietary Pattern 1: High-Protein & Processed Food** | | | | |
| *Aksi Bergizi* non-target schools (n = 253) | 180 (46.9%) | 204 (53.1%) | 1 (Reference) | 1 (Reference) |
| *Aksi Bergizi* target schools (n = 253) | 73 (59.8%) | 49 (40.2%) | 0.59 (0.39, 0.90)* | 0.44 (0.20, 0.99)* |
| **Dietary Pattern 2: Snacks & Sugary Drinks** | | | | |
| *Aksi Bergizi* non-target schools (n = 253) | 226 (59.6%) | 153 (40.4%) | 1 (Reference) | 1 (Reference) |
| *Aksi Bergizi* target schools (n = 253) | 27 (21.3%) | 100 (78.7%) | 5.47 (3.41, 8.77)* | 6.22 (2.68, 14.42)* |
| **Dietary Pattern 3: Healthier Diet** | | | | |
| *Aksi Bergizi* non-target schools (n = 253) | 208 (56.2%) | 162 (43.8%) | 1 (Reference) | 1 (Reference) |
| *Aksi Bergizi* target schools (n = 253) | 45 (33.1%) | 91 (66.9%) | 2.60 (1.72, 3.92)* | 1.73 (0.86, 3.47) |

*Adjusted for sex, age, ethnicity, father's occupation, father's education, mother's occupation, mother's education, household monthly income, main source of information about health behaviors, and dietary self-efficacy.

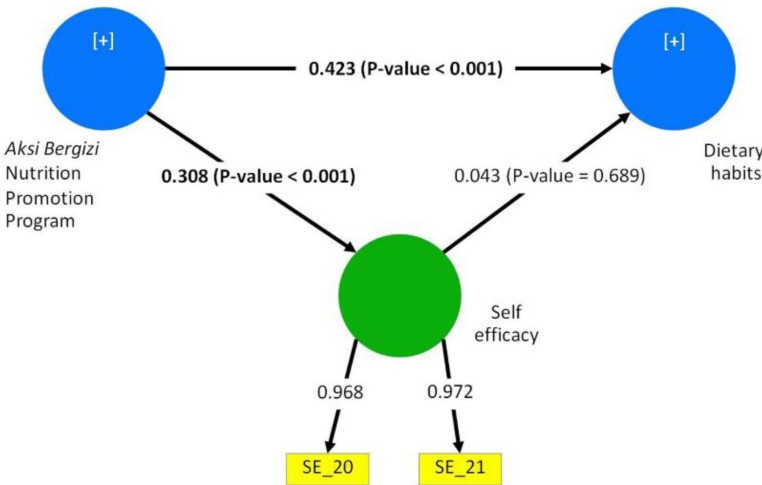

**Fig 2. Structural equation models of mediation of the association between exposure to *Aksi Bergizi* and dietary habits by self-efficacy.**
Legend: SE_20 = Self-Efficacy in Dietary Behaviors; Agree or strongly agree: I am confident that I can limit my consumption of sugary drinks (e.g., soda, energy drinks) to one per week or less. SE_21 = Self-Efficacy in Dietary Behaviors; Agree or strongly agree: I am confident that I can resist the temptation to engage in unhealthy behaviors (e.g., excessive snacking, skipping exercise).

## Conclusion

In this school-based cross-sectional study, we assessed the extent to which dietary behavior of students differed between those in schools that did and did not receive the *Aksi Bergizi* social behavioral change communication (SBCC) program. We found that students in *Aksi Bergizi* target schools were significantly less likely than those in non-target schools to have a High Protein and Processed Foods dietary pattern. Conversely, students in target schools were more likely to adopt a Healthier Diet pattern, though this difference was not statistically significant. However, students in target schools were significantly more likely to follow a Snacks and Sugary Drinks dietary pattern. Regarding mediation, students in *Aksi Bergizi* target schools had significantly different dietary self-efficacy compared to students in non-target schools. However, self-efficacy was not significantly associated with dietary habits. The direct association between exposure to the

*Aksi Bergizi* program and dietary habits remained statistically significant. This suggests that improvements in self-efficacy alone may not be sufficient to drive meaningful changes in students' dietary behaviors. Based on the study findings and the associated issues, we have two main recommendations for future studies and health problems. Firstly, future studies should consider using other measures of dietary intake, such as the 24-hour food recall, to gain additional insights regarding adolescent dietary behaviors. Secondly, future public health programs targeting adolescent dietary behaviors should consider other key behavioral drivers in addition to self-efficacy as the mediator in the behavior change process. The limitations of the study, including the cross-sectional study design (which precludes the ability to make causal inference), the potential information biases, and the limited generalizability, should be considered as caveats in the interpretation of the study findings.

## Supporting information

**S1 File. Data_demo_diet.csv.** S1 - Main data file.
(CSV)

**S2 File. Suppl_data3.csv.** S1 - Data file for SEM.
(CSV)

**S3 File. R1_data_dictionary.xlsx.** S1 - Data dictionary.
(XLSX)

**S4 File. R1_Codes_R 20250322. R.** S1 - R codes.
(R)

**S5 File. R1_suppl_bilingual_questionnaire_INDO_EN.pdf.** Supporting Information – Questionnaire in Bilingual Format.
(PDF)

**S6 File. R1_Inclusivity-in-global-research-questionnaire.docx.** Supporting Information – Inclusivity Questionnaire.
(DOCX)

**S1 Table. R1_supplementary table 1.docx.** Supplementary Table 1 – Frequency of consuming selected food items among participants with dietary patterns identified in factor analysis (n = 506 students).
(DOCX)

## Acknowledgments

We wish to thank all study participants and study school teachers and staff for their valuable time and assistance throughout this study. We also wish to thank our research assistants and data entry staff for their tireless efforts. We would like to thank the Padang Municipality Education Office of the Republic of Indonesia for giving us the permission to conduct this study. This study was part of RDN's thesis, which was completed in partial fulfillment of the requirements for a Master of Science (M.Sc) degree in Epidemiology at Prince of Songkla University, Hat Yai, Songkhla, Thailand.

## Author contributions

**Conceptualization:** Virasakdi Chongsuvivatwong, Wit Wichaidit.

**Data curation:** Ricvan Dana Nindrea, Wit Wichaidit.

**Formal analysis:** Ricvan Dana Nindrea.

**Funding acquisition:** Virasakdi Chongsuvivatwong.

**Investigation:** Sawitri Assanangkornchai, Masrul Muchtar, Virasakdi Chongsuvivatwong, Wit Wichaidit.

**Methodology:** Ricvan Dana Nindrea, Sawitri Assanangkornchai, Masrul Muchtar, Virasakdi Chongsuvivatwong, Wit Wichaidit.

**Project administration:** Ricvan Dana Nindrea, Wit Wichaidit.

**Resources:** Sawitri Assanangkornchai, Masrul Muchtar, Virasakdi Chongsuvivatwong.

**Software:** Virasakdi Chongsuvivatwong, Wit Wichaidit.

**Supervision:** Ricvan Dana Nindrea, Sawitri Assanangkornchai, Masrul Muchtar, Virasakdi Chongsuvivatwong, Wit Wichaidit.

**Validation:** Ricvan Dana Nindrea, Wit Wichaidit.

**Visualization:** Ricvan Dana Nindrea.

**Writing – original draft:** Ricvan Dana Nindrea, Wit Wichaidit.

**Writing – review & editing:** Ricvan Dana Nindrea, Sawitri Assanangkornchai, Masrul Muchtar, Virasakdi Chongsuvivatwong, Wit Wichaidit.

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
