## [Decision Letter · Decision Letter 0]

6 Jan 2025

PONE-D-24-31641Association between receiving the Aksi Bergizi Social Behavioral Change Communication (SBCC) intervention and dietary habits among secondary school students in Padang, IndonesiaPLOS ONE

Dear Dr. Wichaidit,

Thank you for submitting your manuscript to PLOS ONE. After careful consideration, we feel that it has merit but does not fully meet PLOS ONE’s publication criteria as it currently stands. Therefore, we invite you to submit a revised version of the manuscript that addresses the points raised during the review process.

We look forward to receiving your revised manuscript.

Kind regards,

Samuel Adelani Babarinde, PhD

Academic Editor

PLOS ONE

3. In the online submission form, you indicated that [The data underlying the results presented in thie study are available from the corresponding author upon reasonable request.]. All PLOS journals now require all data underlying the findings described in their manuscript to be freely available to other researchers, either 1. In a public repository, 2. Within the manuscript itself, or 3. Uploaded as supplementary information.This policy applies to all data except where public deposition would breach compliance with the protocol approved by your research ethics board. If your data cannot be made publicly available for ethical or legal reasons (e.g., public availability would compromise patient privacy), please explain your reasons on resubmission and your exemption request will be escalated for approval.

Additional Editor Comments (if provided):

Reviewers' comments:

Reviewer's Responses to Questions

**Comments to the Author**

1. Is the manuscript technically sound, and do the data support the conclusions?

Reviewer #1: Yes

Reviewer #2: No

2. Has the statistical analysis been performed appropriately and rigorously? 

Reviewer #1: Yes

Reviewer #2: No

3. Have the authors made all data underlying the findings in their manuscript fully available?

Reviewer #1: Yes

Reviewer #2: Yes

4. Is the manuscript presented in an intelligible fashion and written in standard English?

Reviewer #1: Yes

Reviewer #2: No

5. Review Comments to the Author

Reviewer #1: The questions above promotes the academic standard of the article and meeting the World academic standard. .

Reviewer #2: This article aims to evaluate the impacts and mechanisms of the Aksi Bergizi Program on students' dietary intakes. The following points warrant consideration: 1. Dietary pattern analysis typically relies on intake data of various food groups. It may not be appropriate to utilize binary variables (more than once per week vs. once per week or fewer) for such analysis, as this approach may result in a loss of information. Furthermore, the authors have emerged 22 food groups into only eight, which raises concerns regarding the data resolution. 2. Do the two dietary patterns identified in the manuscript accurately reflect the local dietary habits of Indonesia? The authors may suggest that the fish and fruit dietary pattern is favorable; however, it is essential to determine whether this finding aligns with the original objectives of the Aksi Bergizi Program intervention. 3. The self-efficacy assessment presented in the article appears overly simplistic, consisting of only two questions. It may be beneficial to employ a more comprehensive scale for this evaluation. 4. Before constructing Structural Equation Models (SEMs), a clear theoretical framework or supporting data usually should be established. At least, a comparative analysis of self-efficacy levels between the two groups is needed. 5. The conclusion of the article appears ambiguous. While the SEM analysis indicates that the Aksi Bergizi Program was associated with self-efficacy, it was not associated with dietary habits. Moreover, the pathway between self-efficacy and dietary habits was also found to be insignificant.

6. PLOS authors have the option to publish the peer review history of their article (what does this mean? ). If published, this will include your full peer review and any attached files.

**Do you want your identity to be public for this peer review?** For information about this choice, including consent withdrawal, please see our Privacy Policy .

Reviewer #1: No

Reviewer #2: No

---

## [Author Response · Author response to Decision Letter 1]

31 Mar 2025

Response to the Editor’s Suggestions

COMMENT

In the online submission form, you indicated that [The data underlying the results presented in this study are available from the corresponding author upon reasonable request.]. All PLOS journals now require all data underlying the findings described in their manuscript to be freely available to other researchers, either 1. In a public repository, 2. Within the manuscript itself, or 3. Uploaded as supplementary information.This policy applies to all data except where public deposition would breach compliance with the protocol approved by your research ethics board. If your data cannot be made publicly available for ethical or legal reasons (e.g., public availability would compromise patient privacy), please explain your reasons on resubmission and your exemption request will be escalated for approval.

RESPONSE

We thank the editor for the comment and notification. We will upload the minimal data set necessary to replicate the findings of this study as Supplementary Information along with our codes when we resubmit the manuscript along with the data dictionary. We will also include the English-language translation of our study questionnaire as an additional Supplementary Information section material.

COMMENT

Please include a separate caption for each figure in your manuscript.

RESPONSE

We thank the editor for the comment and have included a separate caption for each figure accordingly

REVIEWER 1

COMMENT

The questions above promotes the academic standard of the article and meeting the World academic standard

RESPONSE

We thank the reviewer for the comments in the Word document and have tried to address them on a point-by-point basis below

Responses to Reviewer 1's comments in the Word document

Abstract

COMMENT

The abstract was poorly written, and Grammarly software is recommended for grammar editing

RESPONSE

We thank the reviewer for the comment and we apologize for the quality of the manuscript. We have used Grammarly to correct the errors accordingly. If the reviewer deems that grammatical mistakes remain in the revised manuscript, please kindly indicate the lines at which they are found in the next iteration and we will make our best efforts to correct them in the subsequent phases.

INTRODUCTION

COMMENT

Paragraph 1: Rewrite the sentence to capture the meaning of the paragraph, and use Grammarly for the grammar editing

RESPONSE

We thank the reviewer for the comment and we have use Grammarly to make corrections as per the reviewer's suggestions.

COMMENT

Paragraph 2: Use Grammarly for grammar editing

RESPONSE

We thank the reviewer for the comment and we have use Grammarly to make corrections as per the reviewer's suggestions.

METHODS

COMMENT

Study design and setting: Rewrite the sentence be formal writing

RESPONSE

We checked our sentence structure and vocabulary use with Grammarly and make revisions to the sub-section accordingly.

COMMENT

Study participants and sample size calculation: Rewrite the paragraph to explain the paragraph clearly and use Grammarly for grammar editing. Indicate the method used in calculating the sample size and the internal consistency of the instruments used

RESPONSE

We thank the reviewer for the comments. We have made the following changes to the latter half of the Study Participants and Sample Size Calculation sub-section as follows:

"Using the sample size calculation formula for comparison of two proportions and procedures for calculation using the R statistical environment assuming 95% level of confidence and 80% power, we obtained the sample size of 253 students per group, thus a total of 506 students."

COMMENT

Section (Exposure: Aksi Bergizi nutrition promotion program): Rewrite ‘’Exposure of Aksi Bergizi nutrition promotion program”; Difficult to express, rewrite to express its meaning; Use Grammarly software for grammar editing

RESPONSE

We thank the reviewer for the comment. The sub-section is meant to describe the exposure, i.e., the Aksi Bergizi nutrition promotion program. Thus, to avoid confusion, we wish to revise the sub-section header to "The Aksi Bergizi nutrition promotion program" with italics on the non-English terms. We have also used Grammarly to revise the manuscript accordingly. Changes made where deemed necessary.

COMMENT

Section (Outcomes: Dietary habits): Change to ‘Dietary habits outcome’; Rewrite the paragraph with formal writing and use Grammarly software for grammar editing

RESPONSE

We thank the reviewer for the comment. The sub-section is meant to describe the measurement of the outcome, i.e., dietary habits of the students in the Aksi Bergizi target and non-target schools. Therefore, we decided to revise the header of the sub-section to "Measurement of dietary habits". We have also used Grammarly to revise the manuscript accordingly. Changes made where deemed necessary.

COMMENT

Section (Mediator: Self-efficacy): Rewrite as ‘Mediator variable: Self-efficacy’; Change to formal writing

RESPONSE

We thank the reviewer for the comment. We decided to revise the header of the sub-section to "Measurement of the mediator (self-efficacy)". We have also used Grammarly to revise the manuscript accordingly. Changes made where deemed necessary.

COMMENT

Sampling methodology: Change to’ research design and sampling’; Rewrite to express the meaning of the paragraph

RESPONSE

We thank the reviewer for the suggestion. The sub-section is meant to describe the sampling procedures, and we have already described the study design at the beginning of the METHODS section. Thus, we wish to suggest an alternative header: "Sampling procedures". We have also revised the paragraph where deemed appropriate as per the reviewer's comments.

COMMENT

Data collection: This is not formal writing, express the paragraph in a formal writing; At this level the authors must be sure of the exact place to drop the questionnaire

RESPONSE

We thank the reviewer for the comments. To improve clarity, we have revised the second half of the sub-section as follows:

"At the end of the data collection period, the students placed their questionnaire in an opaque envelope provided by the investigators, and placed the envelope in a locked box in front of the classroom or at a location otherwise designated by the investigators."

We have also made changes to the other parts of the sub-section as deemed appropriate, including revisions according to Grammarly's suggestions

RESULTS

COMMENT

"Targeted schools had a higher proportion of 14-year-old participants, fathers with an associate's degree...": Table 1 shows that Father has higher Senior high school of 34.0%

RESPONSE

We thank the reviewer for the comment. Just to clarify, our narrations were meant to be a comparison of the characteristics between Aksi Bergizi target schools and non-target schools, and not a description of the participants in the target schoosl only. We have re-checked the findings in Table 1 and wish to hereby respectfully request that we retain the existing descriptions to align with the presented numbers in Table 1.

COMMENT

"Conversely, non-target schools had a higher percentage of fathers working as laborers/manual workers, mothers working as housewives, and mothers with a junior high school education.": This is wrong, the higher percentage was Civil servant/ state enterprise (29.2%); This is wrong, the higher percentage is Senior high school (39.9%)

RESPONSE

We thank the reviewer for the comments. We wish to offer the same response as that to the previous comment

COMMENT

Second paragraph: Rewrite to clearly explain the sentence; This sentence is difficult to express, kindly rewrite it to explain the sentence better; This sentence is confusing, are non-program students the same thing as ‘ non-target schools’?

RESPONSE

We thank the reviewer for the comment. We have tried to make changes accordingly with the help of Grammarly

COMMENT

Third paragraph: Always write with official writing, this no good for academic writing ; Rewrite the discussion to clearly explain the tables and figure 1

RESPONSE

We thank the reviewer for the suggestions and we have made revisions accordingly.

DISCUSSION

COMMENT

The discussion was poorly written without citing an article of similar findings. Rewrite the discussion section to better explain the finding results

RESPONSE

We thank the reviewer for the comment and we have added more citations where we deemed appropriate.

COMMENT

"Future studies should consider using other measures, such as the 24-hour food recall, to gain additional insights [19,20]." This should be discussed under recommendation for further studies

RESPONSE

We thank the reviewer for the comment. The authors have deliberated and felt that the remark addressed the preceding issue that would be more appropriate where it is already located, thus we wish to request that we keep the statement as it appears.

COMMENT

Paragraph three: The article concept is good but poorly expressed, kindly rewrite to clearly discuss the result

RESPONSE

We thank the reviewer for the comment. We have revised the manuscript accordingly and made changes where deemed appropriate.

COMMENT

Paragraph four: This sentence is under limitation of the stud

RESPONSE

We thank the reviewer for the comment. To clarify, the paragraph is meant to communicate the strengths and limitations of the study, with a focus on the latter. Therefore, we wish to keep the content as it appears, but we also welcome more specific suggestions in the next iteration

CONCLUSION

COMMENT

Rewrite the paragraph to clearly explain the paragraph

RESPONSE

We thank the reviewer for the comment. We have revised the manuscript accordingly and made changes where deemed appropriate.

Reviewer 2

1. Dietary pattern analysis typically relies on intake data of various food groups. It may not be appropriate to utilize binary variables (more than once per week vs. once per week or fewer) for such analysis, as this approach may result in a loss of information. Furthermore, the authors have emerged 22 food groups into only eight, which raises concerns regarding the data resolution

RESPONSE

We thank the reviewer for the comment. After deliberations, we agree that using binary variables put us at potential loss of information. We decided that a better (albeit imperfect) approach would be for us to use the reported frequency of consumption as it appeared in the questionnaire, and to use all 22 food groups. We also note that the food frequency questionnaire output values were not meant to be analyzed as discrete numbers, and that errors in the finding regarding patterns of consumption could have resulted. Changes made extensively to the RESULTS section (Tables 2 thru 4). We have identified three dietary patterns instead of two patterns. We have also made changes to the DISCUSSION section according to the revised results.

2. Do the two dietary patterns identified in the manuscript accurately reflect the local dietary habits of Indonesia? The authors may suggest that the fish and fruit dietary pattern is favorable; however, it is essential to determine whether this finding aligns with the original objectives of the Aksi Bergizi Program intervention.

RESPONSE

We thank the reviewer for the comment. To answer the reviewer's question, the investigators deemed that the patterns of dietary habits in the Poultry and Snacks group are consistent with the preferences of adolescents in the region. However, the Fish and Fruits dietary pattern was also consistent with the behaviors that the Aksi Bergizi program was trying to promote.

Because we re-analyzed the data, the findings in the RESULTS section in this iteration of the manuscript differ substantially from those in the previous version. In the DISCUSSION section, we have added remarks regarding consistency between the promoted narrative and behaviors in the Aksi Bergizi program and the observed dietary patterns accordingly.

3. The self-efficacy assessment presented in the article appears overly simplistic, consisting of only two questions. It may be beneficial to employ a more comprehensive scale for this evaluation.

RESPONSE

We thank the reviewer for the comment. Unfortunately, we cannot change the evaluation scale as we have already finished our data collection activity. However, we acknowledge and appreciate the call to explain the measurement of self-efficacy in a more complete manner and have made changes to the METHODS section accordingly.

4. Before constructing Structural Equation Models (SEMs), a clear theoretical framework or supporting data usually should be established. At least, a comparative analysis of self-efficacy levels between the two groups is needed.

RESPONSE

We thank the reviewer for the comment and we have added a short overview of the literature and presented the hypothesis underlying our Structural Equation Model (SEM) in the INTRODUCTION section accordingly.

5. The conclusion of the article appears ambiguous. While the SEM analysis indicates that the Aksi Bergizi Program was associated with self-efficacy, it was not associated with dietary habits. Moreover, the pathway between self-efficacy and dietary habits was also found to be insignificant.

RESPONSE

We thank the reviewer for the comment. When we made the first draft of the manuscript, we noticed the lack of statistical significance in the association between the Aksi Bergizi Program and dietary habits, and between self-efficacy and dietary habits. We acknowledged that our hypothesis was not supported by our findings, but we did not make remarks or discussions extensively. Additional details and remarks added to the DISCUSSION section, including a suggestion that other behavioral drivers should be considered.

---

## [Decision Letter · Decision Letter 1]

15 Jun 2025

PONE-D-24-31641R1Association between receiving the Aksi Bergizi Social Behavioral Change Communication (SBCC) intervention and dietary habits among secondary school students in Padang, IndonesiaPLOS ONE

Dear Dr. Wichaidit,

Thank you for submitting your manuscript to PLOS ONE. After careful consideration, we feel that it has merit but does not fully meet PLOS ONE’s publication criteria as it currently stands. Therefore, we invite you to submit a revised version of the manuscript that addresses the points raised during the review process.

We look forward to receiving your revised manuscript.

Kind regards,

António Raposo

Academic Editor

PLOS ONE

Reviewers' comments:

Reviewer's Responses to Questions

**Comments to the Author**

1. If the authors have adequately addressed your comments raised in a previous round of review and you feel that this manuscript is now acceptable for publication, you may indicate that here to bypass the “Comments to the Author” section, enter your conflict of interest statement in the “Confidential to Editor” section, and submit your "Accept" recommendation.

Reviewer #1: (No Response)

Reviewer #2: (No Response)

Reviewer #3: All comments have been addressed

2. Is the manuscript technically sound, and do the data support the conclusions?

Reviewer #1: Yes

Reviewer #2: Partly

Reviewer #3: Yes

3. Has the statistical analysis been performed appropriately and rigorously? 

Reviewer #1: Yes

Reviewer #2: Yes

Reviewer #3: N/A

4. Have the authors made all data underlying the findings in their manuscript fully available?

Reviewer #1: Yes

Reviewer #2: Yes

Reviewer #3: Yes

5. Is the manuscript presented in an intelligible fashion and written in standard English?

Reviewer #1: Yes

Reviewer #2: No

Reviewer #3: No

6. Review Comments to the Author

Reviewer #1: The authors have improved on the manuscript's corrections; however, before the manuscript's publication, the authors need to make the following corrections:

1. Do grammar editing for the manuscript because some of the sentences were informal

2. The sentences in the red ink under the discussion section should move to the recommendation section.

Reviewer #2: (No Response)

Reviewer #3: Thank you for your submission. Your study addresses an important topic—the association between a school-based SBCC intervention and dietary habits among adolescents in Indonesia. The subject is timely and valuable, particularly in LMICs where adolescent nutrition remains a pressing challenge. I appreciate your efforts in collecting primary data and incorporating structural equation modeling to explore mediation pathways.

That said, several methodological and conceptual aspects need further clarification and improvement to strengthen the manuscript:

The cross-sectional design limits causal interpretation. Please emphasize this more clearly in the abstract and conclusion.

The binary classification of food frequency data reduces the granularity of your dietary behavior measurements. A justification for this decision is warranted, and limitations should be acknowledged more thoroughly.

The self-efficacy scale includes only two items, which weakens the mediation analysis. While you mention this in the response to reviewers, the issue should be more critically discussed in the manuscript.

Your SEM results show that self-efficacy is not significantly associated with dietary habits. This challenges your proposed theoretical model and suggests the need to explore other potential mediators or behavioral determinants.

The finding that students exposed to the Aksi Bergizi program were more likely to follow the "Snacks and Sugary Drinks" pattern is counterintuitive. Please consider offering more context or possible explanations in the discussion section.

In addition, I recommend a final round of language editing to enhance academic clarity and consistency across the manuscript.

7. PLOS authors have the option to publish the peer review history of their article (what does this mean? ). If published, this will include your full peer review and any attached files.

**Do you want your identity to be public for this peer review?** For information about this choice, including consent withdrawal, please see our Privacy Policy .

Reviewer #1: No

Reviewer #2: **Yes: ** Qiang Zhang

Reviewer #3: No

---

## [Author Response · Author response to Decision Letter 2]

9 Jul 2025

Response to the Reviewers’ Comments

REVIEWER 1

COMMENT

The authors have improved on the manuscript's corrections; however, before the manuscript's publication, the authors need to make the following corrections:

RESPONSE

We thank the reviewer for the comment on this iteration. We have made revisions accordingly, particularly by making the sentences more formal and moving the sentences in the DISCUSSION section to the Recommendation part of the CONCLUSION section

COMMENT

1. Do grammar editing for the manuscript because some of the sentences were informal

RESPONSE

We thank the reviewer for the comment. We have tried to make the sentences more formal accordingly.

COMMENT

2. The sentences in the red ink under the discussion section should move to the recommendation section.

RESPONSE

We have moved the sentences in the DISCUSSION section and added the following remarks to the Recommendation part of the CONCLUSION section:

"Based on the study findings and the associated issues, we have two main recommendations for future studies and health problems. Firstly, future studies should consider using other measures of dietary intake, such as the 24-hour food recall, to gain additional insights regarding adolescent dietary behaviors. Secondly, future public health programs targeting adolescent dietary behaviors should consider other key behavioral drivers in addition to self-efficacy as the mediator in the behavior change process."

REVIEWER 2

(No Response)

REVIEWER 3

COMMENT

Thank you for your submission. Your study addresses an important topic—the association between a school-based SBCC intervention and dietary habits among adolescents in Indonesia. The subject is timely and valuable, particularly in LMICs where adolescent nutrition remains a pressing challenge. I appreciate your efforts in collecting primary data and incorporating structural equation modeling to explore mediation pathways.

That said, several methodological and conceptual aspects need further clarification and improvement to strengthen the manuscript:

RESPONSE

We thank the reviewer for the comments and we have tried to revise them accordingly.

COMMENT

The cross-sectional design limits causal interpretation. Please emphasize this more clearly in the abstract and conclusion.

RESPONSE

We thank the reviewer for the comment. We have revised the final sentence in the Conclusion part of the ABSTRACT section to the following:

"However, limitations regarding the cross-sectional design (which precludes the ability to make causal inference), social desirability, and limited generalizability should be considered in the interpretation of the study findings."

We have also included the following remarks in the CONCLUSION section:

"The limitations of the study, including the cross-sectional study design (which precludes the ability to make causal inference), the potential information biases, and the limited generalizability, should be considered as caveats in the interpretation of the study findings."

COMMENT

The binary classification of food frequency data reduces the granularity of your dietary behavior measurements. A justification for this decision is warranted, and limitations should be acknowledged more thoroughly.

RESPONSE

We thank the reviewer for the comment. We have added the following remarks to the DISCUSSION section:

"Furthermore, in our analyses, we dichotomized the responses to the food frequency data as the original answers were ordinal variables with unequal intervals between the categories. Including the responses as integers could have introduced errors during exploratory factor analysis. Thus, we decided to collapse the responses into binary classifications. Such a practice is common in factor analysis and could help to yield clear dietary patterns. However, one key limitation of this process is the loss of details, and the dichotomization as more than once per week vs. once per week or less also implies that there was no seasonality in the consumption of the food items. In the study area, families may consume meat only once per year during religious festivals, but do so in considerable quantity each day for up to one week (e.g., during Hari Raya Eid-Adha), yet the dichotomization presented in our study would not have captured this habit. Incorporating other measurement methods to capture these variations, such as the 24-hour food recall recorded throughout the year, could yield additional insights and enable more detailed measurements."

COMMENT

The self-efficacy scale includes only two items, which weakens the mediation analysis. While you mention this in the response to reviewers, the issue should be more critically discussed in the manuscript.

RESPONSE

We thank the reviewer for the comment. We have added the following remarks to the DISCUSSION section:

"The lack of association between self-efficacy and dietary habits in our SEM thus challenges our proposed theoretical model, and suggests the need to explore other potential mediators in the association between SBCC intervention and dietary behaviors."

We have added the following remarks to the CONCLUSION section:

"…future public health programs targeting adolescent dietary behaviors should consider other key behavioral drivers in addition to self-efficacy as the mediator in the behavior change process."

COMMENT

Your SEM results show that self-efficacy is not significantly associated with dietary habits. This challenges your proposed theoretical model and suggests the need to explore other potential mediators or behavioral determinants.

RESPONSE

We thank the reviewer for the comment. We have added the following remarks to the DISCUSSION section:

"The lack of association between self-efficacy and dietary habits in our SEM thus challenges our proposed theoretical model, and suggests the need to explore other potential mediators in the association between SBCC intervention and dietary behaviors."

We have also added the following remarks to the CONCLUSION section:

"...future public health programs targeting adolescent dietary behaviors should consider other key behavioral drivers in addition to self-efficacy as the mediator in the behavior change process."

COMMENT

The finding that students exposed to the Aksi Bergizi program were more likely to follow the "Snacks and Sugary Drinks" pattern is counterintuitive. Please consider offering more context or possible explanations in the discussion section.

RESPONSE

We thank the reviewer for the comment. We have added the following remarks to the DISCUSSION section accordingly:

"This unexpected change was not found in a similar study in another region of Indonesia. Another study showed that a school in Indonesia with on-site food service with a nutritionist had better nutrient adequacy among students than a school that had catering without a nutritionist. Thus, it is possible that the lack of a catering component with nutritionist involvement could explain for parts of the observed unhealthy dietary patterns in our study."

COMMENT

In addition, I recommend a final round of language editing to enhance academic clarity and consistency across the manuscript.

RESPONSE

We thank the reviewer for the suggestion. We have made our best attempt to edit our writing to improve clarity and consistency accordingly.

---

## [Decision Letter · Decision Letter 2]

4 Aug 2025

PONE-D-24-31641R2Association between receiving the Aksi Bergizi Social Behavioral Change Communication (SBCC) intervention and dietary habits among secondary school students in Padang, IndonesiaPLOS ONE

Dear Dr. Wichaidit,

Thank you for submitting your manuscript to PLOS ONE. After careful consideration, we feel that it has merit but does not fully meet PLOS ONE’s publication criteria as it currently stands. Therefore, we invite you to submit a revised version of the manuscript that addresses the points raised during the review process.

We look forward to receiving your revised manuscript.

Kind regards,

António Raposo

Academic Editor

PLOS ONE

Journal Requirements:

Reviewers' comments:

Reviewer's Responses to Questions

**Comments to the Author**

1. If the authors have adequately addressed your comments raised in a previous round of review and you feel that this manuscript is now acceptable for publication, you may indicate that here to bypass the “Comments to the Author” section, enter your conflict of interest statement in the “Confidential to Editor” section, and submit your "Accept" recommendation.

Reviewer #2: (No Response)

Reviewer #3: All comments have been addressed

2. Is the manuscript technically sound, and do the data support the conclusions?

Reviewer #2: Partly

Reviewer #3: Yes

3. Has the statistical analysis been performed appropriately and rigorously? 

Reviewer #2: I Don't Know

Reviewer #3: Yes

4. Have the authors made all data underlying the findings in their manuscript fully available?

Reviewer #2: Yes

Reviewer #3: Yes

5. Is the manuscript presented in an intelligible fashion and written in standard English?

Reviewer #2: Yes

Reviewer #3: Yes

6. Review Comments to the Author

Reviewer #2: (No Response)

Reviewer #3: The authors have carefully addressed all the comments and suggestions raised during the previous round of review. The revised manuscript demonstrates significant improvements in clarity, methodology, and presentation. All major and minor concerns have been adequately resolved, and no further revisions are necessary from my perspective.

Therefore, I find the current version of the manuscript suitable for publication in its present form.

7. PLOS authors have the option to publish the peer review history of their article (what does this mean? ). If published, this will include your full peer review and any attached files.

**Do you want your identity to be public for this peer review?** For information about this choice, including consent withdrawal, please see our Privacy Policy .

Reviewer #2: **Yes: ** Qiang Zhang

Reviewer #3: No

---

## [Author Response · Author response to Decision Letter 3]

5 Aug 2025

Response to the Reviewers’ Comments

JOURNAL REQUIREMENTS

COMMENT

RESPONSE

We thank the Editor for communicating the requirement. To the best of our knowledge, there are no retracted papers in our citations.

REVIEWER 1

COMMENT

(Attachment)

RESPONSE

The attachment with Reviewer 1’s comments appeared to be from the first round of review (dated 19 November 2024 and submitted to the journal on 20 November 2024), and the commented document appeared to be our original submission. We have made changes to the manuscript based on these comments when we submitted the first revision (R1). Thus, we decided to refrain from addressing the comments in Reviewer 1’s attached document

REVIEWER 2

COMMENT

(No Response)

RESPONSE

(No Response)

REVIEWER 3

COMMENT

The authors have carefully addressed all the comments and suggestions raised during the previous round of review. The revised manuscript demonstrates significant improvements in clarity, methodology, and presentation. All major and minor concerns have been adequately resolved, and no further revisions are necessary from my perspective.

Therefore, I find the current version of the manuscript suitable for publication in its present form.

RESPONSE

We thank the reviewer for the comment, and confirm keeping the manuscript as it appears.

---

## [Editor Report · Decision Letter 3]

14 Aug 2025

Association between receiving the Aksi Bergizi Social Behavioral Change Communication (SBCC) intervention and dietary habits among secondary school students in Padang, Indonesia

PONE-D-24-31641R3

Dear Dr. Wichaidit,

We’re pleased to inform you that your manuscript has been judged scientifically suitable for publication and will be formally accepted for publication once it meets all outstanding technical requirements.

Kind regards,

António Raposo

Academic Editor

PLOS ONE
---

## [Editor Report · Acceptance letter]

PONE-D-24-31641R3

PLOS ONE

Dear Dr. Wichaidit,

I'm pleased to inform you that your manuscript has been deemed suitable for publication in PLOS ONE. Congratulations! Your manuscript is now being handed over to our production team.

Kind regards,

on behalf of

Dr. António Raposo

Academic Editor

PLOS ONE